# Recent Strategies for the Remediation of Textile Dyes from Wastewater: A Systematic Review

**DOI:** 10.3390/toxics11110940

**Published:** 2023-11-19

**Authors:** Manikant Tripathi, Sakshi Singh, Sukriti Pathak, Jahnvi Kasaudhan, Aditi Mishra, Saroj Bala, Diksha Garg, Ranjan Singh, Pankaj Singh, Pradeep Kumar Singh, Awadhesh Kumar Shukla, Neelam Pathak

**Affiliations:** 1Biotechnology Program, Dr. Rammanohar Lohia Avadh University, Ayodhya 224001, India; 2Department of Microbiology, Punjab Agricultural University, Ludhiana 141001, India; 3Department of Microbiology, Dr. Rammanohar Lohia Avadh University, Ayodhya 224001, India; 4Department of Biochemistry, Dr. Rammanohar Lohia Avadh University, Ayodhya 224001, India; 5Department of Botany, K.S. Saket P.G. College, Ayodhya 224001, India

**Keywords:** bioremediation, dye degradation, environmental pollutants, genetic engineering, microbial fuel cells, nanotechnology, textile wastewater treatment

## Abstract

The presence of dye in wastewater causes substantial threats to the environment, and has negative impacts not only on human health but also on the health of other organisms that are part of the ecosystem. Because of the increase in textile manufacturing, the inhabitants of the area, along with other species, are subjected to the potentially hazardous consequences of wastewater discharge from textile and industrial manufacturing. Different types of dyes emanating from textile wastewater have adverse effects on the aquatic environment. Various methods including physical, chemical, and biological strategies are applied in order to reduce the amount of dye pollution in the environment. The development of economical, ecologically acceptable, and efficient strategies for treating dye-containing wastewater is necessary. It has been shown that microbial communities have significant potential for the remediation of hazardous dyes in an environmentally friendly manner. In order to improve the efficacy of dye remediation, numerous cutting-edge strategies, including those based on nanotechnology, microbial biosorbents, bioreactor technology, microbial fuel cells, and genetic engineering, have been utilized. This article addresses the latest developments in physical, chemical, eco-friendly biological and advanced strategies for the efficient mitigation of dye pollution in the environment, along with the related challenges.

## 1. Introduction

Water is one of the most essential factors in our lives. It has been reported that approximately 71% of our planet is covered by water, consisting of 97.5% seawater and only 2.5% freshwater [1]. Industrial wastewater is very harmful for our environment and causes various adverse impacts on the ecosystem. Most wastewater is generated from the textile, cosmetics, printing, paper, and rubber industries [2]. Dyes are categorized based on their features and applications [3,4]. The textile industry generates huge amounts of highly toxic chemicals that are released at various stages of processing [5]. In addition, it is well documented that textile dyes, without proper treatment, pose serious eco-toxicological threats to living forms [6,7].

Textile dye-containing wastewater is also one of the major sources of water pollution, and it affects the environment by disturbing aquatic life, inhibiting the photosynthesis process in aquatic plants, and affects human health by causing breathing difficulties, irregular heartbeats, skin rashes, dizziness, and cancer [4,6,8]. Different types of dyes are used in the textile industry, such as sulphur dye, azo dye, acid dye, and basic dye. Most of the common dyes applied in the textile industry are called azo dyes [9,10].

Since ancient times, mankind has known about and utilized dyes [11]. In many manufacturing sectors, including the paper industry and many more, a number of techniques are effective in removing dyes [12,13,14]. It has been found that the dyes used in the textile industry have high toxicity and a potential carcinogenic nature [8,15]. The textile industry is responsible for a broad range of impacts on the environment [16]. Textile dyes can cause diseases ranging from dermatitis to central nervous system disorders [17]. Consequently, treating toxic wastewater containing different types of textile dyes is a very compelling issue currently [18,19,20]. Accommodating a single or multiple azo groups, azo dyes constitute 60–70% of known dye structures [21,22,23].

The treatment of dye-containing wastewater occurs through various methods like physical, chemical, biological, and recent advanced technologies such as using nanotechnology, microbial biosorbents, microbial films, genetic engineering, plant–microbe-mediated techniques, and others [4,14]. In recent studies, researchers have reported the use of chemical-based methods for the remediation of pollutants from wastewater [24,25,26]. Metal-based organic frameworks are also used as sensitive and selective sensors for detecting hazardous pollutants [27]. However, physical and chemical methods are not cost-effective [6].

Bioremediation is an eco-friendly and cost-effective option to remove various types of organic and inorganic toxicants from polluted environments [28]. Microbial processes using algae, bacteria, yeast, and fungi are low-cost and can be successfully utilized for dye remediation. Microorganisms are capable of breaking down the azo bonds present in dye molecules through their enzymatic activity [6,29]. Advanced treatment strategies using microorganisms can make the remediation process more effective, while appropriate environmental conditions should also be ensured for effective bioremediation [30].

Researchers have reported that non-treated wastewater has negative effects on ecosystems and raises health concerns among humans [4]. Therefore, it is important to treat dye-containing wastewater effectively with low-cost and environmentally friendly approaches to avoid the dangers of such pollutants. This review systematically addresses different technologies such as microbial mechanisms, nanotechnology, bioreactors, microbial biofilm, phytoremediation methods, challenges with these methods, and how advanced strategies can be effectively operated to clean dye-containing wastewater contributing to a greener and cleaner environment.

## 2. Data Collection and Bibliometric Analyses

In the present study, we searched for “textile waste water treatment”, “nanotechnology”, “genetic engineering”, “bioreactors,”, “bioremediation”, “microbial fuel cells,” and “dye degradation” in the Web of Science using VOS viewer 1.6.18. The number of publications and citations relating to the selected term displayed a peak in the current year. A network visualization map indicating the co-citation of the most-cited terms is presented in Figure 1. Most recent research has paid attention to the bioremediation of textile dyes from wastewater, and the depictions reveal that bioremediation, nanotechnology, and genetic engineering are among the most studied areas in the structure of this study.

## 3. Dyes Used in Textile Industry

Azo dyes are frequently used in a range of industries, including those that produce food, medicines, paper, cosmetics, textiles, leather, and other products [6,31]. In one study, Benkhaya et al. [31] discussed the classification and characteristics of dyes. They reported the chemical classification of azo dyes and their structural properties. Several researchers reported the different types of dyes that are used in the textile industry, such as acidic, basic, direct, sulphur and azo dyes, along with their applications [31,32,33,34,35,36].

## 4. Approaches for Dyes Remediation

Pollutants such as dyes from the textile industry are present in wastewater. Various techniques, such as physical, chemical, and biological methods, are used for the decontamination of dyes in aquatic environments. The physical and chemical procedures used to treat textile wastewater are inadequate and not always environmentally feasible. A safe method to detoxify dyes from wastewater is bioremediation technology using microorganisms or plants. Nowadays, advanced technologies such as bioremediation, including nanotechnology, bioreactors, microbial fuel cells, genetic engineering, and others, are being applied in the treatment of dye pollution in the environment (Figure 2).

Different approaches to remove various types of dye from the environment are critically described below.

### 4.1. Physical and Chemical Approaches

Adsorption, ion exchange, and membrane filtration are a few of the physical methods used in the remediation of environmental pollutants. Some of these physical techniques are discussed below.

#### 4.1.1. Adsorption

Adsorption is used for the management of industrial wastewater. Adsorption is a surface-based phenomenon in which the solid surface of the adsorbent attracts charged ions or molecules, which then adsorb onto the surface. Several types of forces are responsible for dye molecule adsorption, including hydrophobic, electrostatic interaction, hydrogen bonding, and van der Waals forces [37]. The adsorption process removes dyes from contaminated wastewater, and there is a possibility of upcycling the adsorbent for reuse in treatment [3]. The phenomenon of adsorption is dependent on the adsorbent, which contains pore-like structures that are required for the quick and systematic adsorption of dye molecules from wastewater [38,39]. Most adsorbents, such as silica gel, alumina, zeolite, and activated carbon, are commonly used for the removal of toxic dyes from contaminated wastewater [40]. Other mechanisms like complexation and ion exchange are applied in the remediation of dyes [41,42,43]. In their study, Briao et al. [44] found that ZSM-5 zeolite adsorbent is used in the treatment of dye-containing wastewater, such as basic fuchsin, crystal violet, and methylene blue, with a degradation percentage of 81.2%, 75.3%, and 86.6%, respectively. In another study, Madan et al. [45] reported 90% Congo red decolorization by ZnO, used as an adsorbent, while Harja et al. [46] reported Congo red dye remediation with the help of fly ash generated from a local powerplant. Several researchers have also reported different types of organic adsorbents and their composites for dye remediation from aqueous solutions, and some of the most interesting adsorbents are conducting polymers in their different forms, including powders and aerogels, as well as biopolymers [47,48,49].

#### 4.1.2. Ion-Exchange Method

In the ion-exchange method, effective separation is accomplished by creating a complex bond between resin, which is a bed reactor, and a solute. Akpomie and Conardie [3] and Ahmad et al. [50] found that the process of dye removal in ion exchange mainly depends on strong interactions between charged molecules in the dye and the functional group of the resin. The cation exchangers and anion exchangers are used as resins to separate solutes with different surface charges [51]. Many researchers reported that dyes like acid orange 10 are removed by an anion-exchanger resin named Amberlite IRA-400, and the percentage of dye removed was 96.8% [52]. Another dye, acid black, was remediated at a rate of 100% by an anion exchanger that was synthesized using cellulose [53]. Disperse violet 28 dye is generally removed by cation-exchanger resin, and the dye removal percentage is 91.7 [54].

#### 4.1.3. Membrane Filtration

Membrane filtration is one of the most important physical methods for textile dye removal from wastewater [55,56]. In this method, due to the small pore size of the membrane, molecules larger than the filter pores become trapped. Microfiltration is a membrane-based phenomenon that involves the separation of dyes in the size range of 0.1–0.10 μm [57]. In this process, the waste materials or dyes are remediated from the liquid with the help of a microporous membrane. Ultrafiltration is another membrane-based method. Collivignarelli et al. [58] reported that dye color removal capabilities are acquired from wastewater with the help of ultrafiltration. The reactive black dye solution uses an ultrafiltration ceramic membrane and decolorizes the dye at different concentrations [59]. Reverse osmosis is also a membrane filtration mechanism that is used for the treatment of industrial wastewater that contains dyes [60,61]. Several researchers have reported the application of filtration technology for the treatment of wastewater containing dyes [62,63,64].

#### 4.1.4. Fenton Process

The two major approaches for the degradation of dyes present in textile water are Fenton and photo-Fenton [65,66]. These processes are carried out by using a Fenton reagent, namely H_2_O_2_, and a soluble iron (II) salt mixture [67,68,69]. In a study by Ledakowicz et al. [70], it was reported that the degradation of three dyes (add red 27, reactive blue 81, and add blue 62) occurred with the use of a Fenton reagent. They concluded that this reaction is very simple and fast. In another study, Chen et al. [71] used the stopped flow technique to study the degradation of methylene blue and rhodamine B with Fenton reagent. Zawadzki and Deska [72] published a review concerning the degradation of dyes by combining advanced oxidation processes with different methods, such as utilizing ozone, hydrogen peroxide, and persulfate to degrade rhodamine B. They concluded that degradation is achieved by using UV in a photo-assisted ozonation, which was the most effective method among all of the techniques tested.

#### 4.1.5. Ozonation

Dye remediation using ozonation is another important treatment technology. Shriram and Kanmani [73] concluded in their study that H_2_O_2_, UV radiation, etc., are used in the ozonation process. They provided a detailed study about the mechanism, influencing factors, and initiators of ozonation. Venkatesh and colleagues [74] reported dye remediation by means of combined ozonation and anaerobic treatment strategies. They reported that the cost of ozonation can be reduced using an upflow anaerobic sludge blanket reactor. In another study, Cardoso et al. [75] reported nanomaterial-based catalysts for the ozonation process in the remediation of dyes. They used copper (II)-doped carbon dots as catalysts in catalytic ozonation. They also analyzed the degradation of four dyes, namely methyl orange, orange II sodium salt, reactive black, and remazol brilliant blue R (RBB-R). In a recent research work, Lanzetta et al. [76] reported using the ozonation process for color remediation from tanning wastewater. Further research is needed on the decolorization of wastewater using ozonation.

### 4.2. Biological Approaches

The physical and chemical techniques used in dye decolorization are costly. Traditional biological techniques are used exclusively or with chemical and physical techniques for dye decolorization. Nonetheless, advanced biological methods that are less expensive and more effective with lesser secondary sludge production are emphasized [13,28,77]. Microbial methods are effective for the bioremediation of different types of organic and inorganic pollutants [30,78,79]. On the basis of different research findings, for the decolorization of dyes, bacterial treatment serves as an efficient strategy [80]. Figure 3 presents various biological methods for the bioremediation of dyes from contaminated environments.

#### 4.2.1. Enzymatic Method

Bioremediation by enzymes is an ingenious, favorable, environmentally friendly technique [81]. The enzymatic degradation method consists of finding the attributes of microbes or genetically modified microbes, designing enzymes to metabolize the dyes, and transforming the harmful form of dyes into harmless forms or non-toxic forms [81]. Regarding azo dye degradation, a number of studies have been conducted to understand different type of enzymatic activities that help in the degradation of toxic dyes [81,82]. A class of enzyme, azoreductase, has been described by Mendes et al. [83] as carrying out the reduction reaction causing the breakdown of azo bonds (-N=N-) present in dyes, and converting the aromatic amine into colorless water. The enzyme laccase can be used in the treatment of different toxic textile dyes [84]. Lignin and Mn peroxidase have been widely studied; peroxidase enzymes have been used in the degradation of toxic textile dyes [85]. For the bioremediation of hydrocarbons and pesticides, enzymes produced by aerobic bacteria such as *Alcaligene* sp. and *Pseudomonas* sp. are also used [81]. Enzymes produced by different types of fungi, such as lignin peroxidases, azoreductases, and laccases from white rot fungi, can also take part in the biodegradation of textile dyes [86]. In a recently published review, Pinheiro et al. [87] showed the role of different enzymes in the bioremediation of dyes (Figure 4).

Various researchers such as Shahid et al. [88] found that the strain MR-1/2 of multifarious *Bacillus* species efficiently decolorized dyes such as reactive black-5, reactive red-120, direct blue-1, and Congo red, which additionally helped in the growth of the mung bean plant by alleviating azo dye toxicity. Meanwhile, Vineh et al. [89] found 100% decolorization of most of the dyes used in the study at pH 7, 25 °C, for 60 min by using peroxidase immobilized on a functionalized reduced grapheme oxide. In another study, Navas et al. [90] reported 20–100% decolorization of dyes at pH 5–9 using laccase, which was purified and extracted from the thermophilic *Thermus* species. In another study, Kalsoom et al. [91] found 95% degradation of remazol turquoise blue 133G dye with peroxidase from *Brassica oleracea*, while Gao et al. [92] achieved 72–80% decolorization of the azo dyes reactive blue 19 and acid orange 7 at a neutral to alkaline pH at relatively high temperatures using laccase enzymes immobilized in vault nanoparticles.

#### 4.2.2. Microbial Remediation

Several microbes, such as bacteria, fungi, yeast, algae, and actinomycetes, are utilized for treating textile dyes from aquatic environments [6,10,93,94,95,96,97].

##### Bacterial Remediation

Among all the groups of microbes, decolorization by bacteria is significant. From a biotechnological perspective, bacteria offer many advantages as they contain abundant degradative enzymes, consequently having the capacity to degrade dyes of a broad range [6,98,99]. The basic advantage of dealing with bacteria is their efficiency in growing quickly, and their ability to be cultured easily. The organic pollutants that are aromatic hydrocarbon-based and chlorinated can be catabolized by bacteria as their source of energy [100]. Bacteria also have the capability to oxidize textile dyes based on sulphur to H_2_SO_4_ [101].

##### Pure Bacterial Cultures

Using a pure bacterial culture with one type of bacteria, the biodecolorization of dyes has been reported in several studies. In one investigation, Louati et al. [102], found 100% decolorization of dyes at pH 8 by using *Pseudomonas aeruginosa* strain Gb30, whereas Montanez-Barragan et al. [103] observed above 90% decolorization of dyes at pH 6–11 with *Halomonas* species. Another researcher, Shi et al. [104], found 100% dye removal of Brilliant Crocein by using the bacteria *Providencia rettgeri*, while Fareed et al. [105] found 80–100% decolorization of dyes by using free and immobilized cells of *Bacillus cereus* at temperatures of 32 °C, 37 °C, and 45 °C. Srinivasan et al. [106] achieved 88.35–96.30% decolorization of different azo dyes, while Du et al. [107] observed the complete decolorization of malachite green and crystal violet at pH 3–10 and a temperature of 20–45 °C after 12 h of incubation under optimal environmental conditions using *Serratia* species, which is a new bacterial strain identified via 16S rDNA analysis. In a recent study, Tripathi et al. [10] observed 98% dye decolorization of crystal violet dye using a native multiple metal-tolerant *Aeromonas caviae* MT-1 isolate.

##### Mixed Bacterial Cultures

For decolorizing diverse groups of dyes, bacteria of a single species are not efficient enough for their remediation, and this is one of the most prominent challenges for environmental biotechnologists [108]. Many researchers have worked on a consortium or mixed bacterial culture for the degradation of dyes. In a study, Ayed et al. [109] observed 90% dye decolorization at 35 °C using a bacterial consortium consisting of *Sphingomonas paucimobilis*, *Pseudomonas putida*, and *Lactobacillus acidophilus*. In another study, Guo et al. [110] found 93% decolorization at 40 °C and pH 10 of Methanil Yellow G dye using a consortium with *Halomonas, Marinobacter*, and *Clostridii salibacter*., Meanwhile, in another investigation, Joshi et al. [108] reported dye decolorization of 24–94% using a consortium of six bacterial strains: *Pseudomonas stutzeri* AK1, *P. stutzeri* AK2, *P. stutzeri* AK3, *Bacillus* sp. AK4, *P. stutzeri* AK5, and *P. stutzeri* AK6. Likewise, Bera et al. [111] reported 85% acid orange dye decolorization after nearly 23 h, with yeast as a supplementary source and a bacterial consortium called novel bacterial consortium SPB92 composed of four bacterial strains, i.e., *Pseudomonas stutzeri* (MW219251), *Bacillus tequilensis* (MW110471), *B. flexus*. (MW13 flexus and *Kocuria rosea* (MW 132411), whereas Neihsial et al. [112] found 85–97% degradation of dyes using a consortium including bacteria of different genera, like *Acinetobacter, Comamonas, Trichococcus, Erwinia, Dysgonomonas,* and *Citrobacter*. In another study, Barathi et al. [113] found that the bacterial consortium with three bacterial species, *B. subtilis, Brevibacillus borstelensis,* and *B. firmus*, was able to degrade dye at a better rate at lower concentrations of dye, but the ability of degradation decreased with elevated concentrations of dye. Several bacterial isolates have been reported for dye decolorization (Table 1).

##### Actinomycetes

Microorganisms, particularly actinomycetes, play a significant role in the decolorization of dyes. Several researchers have reported the decolorization of dyes using actinomycetes. Zhou and Zimmermann [97] observed 3–10% decolorization of dyes by *Streptomyces* species. On the other hand, in recent research, Dong et al. [119] observed 99% decolorization of dyes using *Streptomyces* sp. S27. In another study, Adenan et al. [120] observed 64–94.7% decolorization of different dyes using actinomycetes. They reported the decolorization of triphenylmethane dyes using *Streptomyces bacillaris* through biosorption and biodegradation mechanisms. Raja et al. [121] studied the decolorization of amido black dye by means of actinomycetes isolated from marine sediments under aerobic conditions. They found significant 88% decolorization at a 5 ppm dye concentration. Meanwhile, Blánquez et al. [122] observed decolorization of 6–70% using the actinomycetes *Streptomyces ipomoeae* CECT 3341. In another study, Kameche et al. [123] studied the decolorization of the azo dye Evans blue using four strains of *Streptomyces* isolated from soils. They observed a 97% remediation rate at an initial 50 mg/L Evans blue concentration.

##### Phycoremediation

Studies suggest that azo dyes are utilized as a carbon and energy source for algae, which are then degraded into aromatic amines, subsequently being converted into simple inorganic and organic compounds [124]. For the investigation of dye decolorization from textile wastewater samples, *Chlorella* species, *Oscillatoria* species, *Phormidium* species, and *Synechocystis* have been widely used [125]. Many functional groups like carboxy, carbonyl, hydroxy, phosphoryl, and amide groups are present in algal cell walls, which help in dye decolorization [126]. In another investigation, Mahajan et al. [127] reported 70–100% decolorization of methyl red dye at pH 3.5–9.5 using *Chara vulgaris* L. In another study by Boulkhessaim et al. [128], they reported 45–80% decolorization of dyes using *Chlorella vulgaris*, whereas Dellamatrice et al. [129] observed 91% dye decolorization using *Cyanobacterium phormidium*. In another study, Alprol et al. [130] found 75.7% and 61.11% decolorization of dyes by using *Arthrospira platensis* complete dry biomass and lipid-free biomass. Meanwhile, Mansour et al. [131] reported 93% decolorization of methylene blue using *A. platensis.* These studies indicate that algae application for effective remediation of dye pollution is a viable and eco-friendly option for a sustainable and green environment.

##### Yeast-Mediated Dye Decolorization

Yeast has not been as thoroughly studied and used in the decolorization of dyes as filamentous fungi and bacteria [132]. The removal of dyes using yeast was reported in the biosorption process [132]. The decolorization of azo dye through yeasts is achieved using the enzyme azoreductase present in yeast [133]. *Galactomyces geotrichum* MTCC1360 has the capacity to decolorize azo dyes [134]. Researchers like Guo et al. [135] reported 92% decolorization of the azo dye Acid Scarlet GR using the newly isolated salt-tolerant yeast strain *Galactomyces geotrichum* GG. In another study, Ali et al. [136] reported 82% decolorization of lignin-like dyes and wastewater containing textile dyes using a recently formed oleaginous yeast consortium with three yeast cultures: *Yarrowia* sp. SSA1642, *Barnettozyma californica* SSA1518, and *Sterigmatomyces halophilus* SSA1511.

##### Phytoremediation

Phytoremediation involves the plant-mediated treatment of contaminants from the environment, and it is a cost-effective solution for the clean-up of various types of pollutants [137,138]. This process operates through several mechanisms or processes to remediate contaminants. The different phytoremediation strategies, such as phytoextraction (uptake or absorption of contaminants by roots into the shoots for metabolization), phytostabilization (compounds secreted by the plant immobilize contaminants rather than degrade them), phytovolatilization (involves translocation of contaminants by roots to aerial plant parts where they volatilize into the atmosphere), and rhizofiltration (plant roots absorb the contaminants, which are then metabolized or stored), are widely used, and accepted as a cost-effective environmental restoration technology [139,140,141,142].

In phytoremediation, plants interact at the physical, chemical, biological, and microbial levels to reduce pollutant toxicity. This employs a variety of processes depending on the form and quantity of the pollutant [140]. In their investigation, Biju et al. [143] found 75 ± 0.5% and 82 ± 0.5% decolorization of a mixture of azo dyes using *Salvinia* species. Rane et al. [144] observed complete decolorization of sulfonated remazol red dye and effluents of the textile industry using *Alternanthera philoxeroides*. In another study, Imron et al. [145] found the decolorization of 80.56 ± 0.44% of methylene blue dye using duckweed (*Lemna minor*), whereas Baldawi et al. [146] observed 85% decolorization of methylene blue dye using the floating plant *Azolla pinnata*.

## 5. Recent Strategies for Remediation of Dyes

Nowadays, advanced methods using microbial fuel cells, nanotechnology, and others are showing promise for the treatment of dye pollution.

### 5.1. Genetically Engineered Microorganisms (GEMs)

Through genetically altered microbial strains, bioaugmentation has given a solution for effective remediation of pollutants from environment [147,148]. Genetic engineering is specifically applied to transformed microbes (like bacteria, fungi, and yeast) using molecular tools. GEMs can be employed for the bioremediation of environmental pollutants [148,149,150]. In their study, Bu et al. [151] performed dye decolorization experiments with mutant laccase Lacep69, D500G, and wild-type laccase. They found that the decolorization rate of D500G was better than that of wild-type laccase, with 78% decolorization for acid violet. In another study, Dixit and Garg [152] transferred the azoreductase enzyme-coding gene azoK from the bacteria *Klebsiella pneumonia* to *Escherichia coli* for successful 95% decolorization of azo dyes in less than 24 h. Meanwhile, Chang et al. [153] reported a 10% increase in the biodecolorization of azo dyes by engineering the *E. coli* genes coding for azoreductase present in Pseudomonas luteola.

### 5.2. Microbial Biosorbents

Biosorption is an energy-independent process, where sorption occurs with the help of biological molecules [154]. From water or soil, microbial biosorbents absorb or adsorb xenobiotic compounds such as carcinogens, drugs, metals, dyes, etc. [14,155]. In their investigation, Bayat et al. [156] reported the removal of 97% dye by using *Chromochloris zofingiensis* as a potential dye biosorbent. In another study, Sun et al. [157] reported a remediation efficiency of 92.2% at pH 2 by using the marine algae *Enteromorpha prolifera* for the biosorption of Direct Fast Scarlet 4BS from an aqueous solution. Silva et al. [158] observed a maximum biosorption capacity of 76.3359 mg/g by using natural biosorbents made from weeds for the biosorption of methylene blue dye. In another study, Alarcón et al. [159] reported 95% dye removal from wastewater from the textile dye industry at pH 2.6 by using orange seed powder as a suitable biosorbent. In a recently published review, Tripathi et al. [14] discussed the importance of microbial biosorbents for the bioremediation of heavy metals and dye-contaminated environments.

### 5.3. Bioreactors

Over the past two decades, membrane bioreactors have shown promising and efficient potential for treating contaminated wastewater [160] A researcher group, Manzoor et al. [161], reported 99% removal of dyes using a modified anaerobic forward osmosis membrane bioreactor (AnMBR) process. In another study, Jin et al. [162] also reported 99% removal of navy blue dye using modified AnMBR, while Castro et al. [163] reported 61% removal of reactive orange 16 using AnMBR treatment. In another investigation, Zhang et al. [164] found >90% dye removal of most azo dyes using an anaerobic flat-sheet ceramic membrane bioreactor. Bioreactor systems can be applied at a larger level for the abatement of dye pollution in ecosystems.

### 5.4. Nanoparticles Based Bioremediation

Dye and the degradation byproducts can have an adverse impact on the environment and people’s health. Traditional methods of dye remediation are less effective and more expensive. In recent years, nano-bioremediation has emerged as a viable alternative for the effective and environmentally friendly removal of dyes from the environment [165]. Numerous industries, including agriculture, energy, medicine, and the environment, all use nanotechnology [14,166,167]. The development of nanoparticles by environmentally friendly microbes is helpful for the nano-bioremediation of contaminants. Nanoparticles (NPs) have distinctive structural properties, which make them attractive for the treatment of textile wastewater. These NPs act as adsorbents for the successful removal of dyes. In their study, Balrabe et al. [165] reported that eosin yellowish displayed efficient removal (95%) at 75 ppm using Fe_2_O_3_ nanoparticles, pH 6, with a 15 min exposure time. In another study, Darwesh et al. [168] reported the *F. oxysporum* OSF18 strain-synthesized CuO-NPs immobilized in alginate beads, which exhibited 90% dye remediation efficiency. Functionalized nanomaterials may also be applied in dye removal. Rani and Shanker [169] reported the use of functionalized nanomaterials for the remediation of organic dyes. Further research is necessary to study the role of nanomaterials for large-scale in situ bioremediation.

### 5.5. Microbial Fuel Cells (MFCs)

Microbial fuel cells (MFC) are employed as oxidizing agents that oxidize organics in the anode and transfer electrons to reduce the terminal electron acceptor oxygen to the cathode electrode. The whole system setup includes electrode materials, external resistance and applied potentials, the initial level of azo dyes, and co-substrates [170]. In MFCs, microorganisms are used to generate bioelectricity [171]. MFCs can be characterized as (1) single-chambered and (2) double-standard. Double-standard chambers are separated by a proton exchange membrane. Substrates are oxidized anaerobically in the anode chamber by microbes to produce electrons. In their investigation, Jain and Zhen [172] reported a bio-electrochemical system treatment in which electrons (acceptors and donors) interact to achieve maximum dye remediation in contaminated wastewater by using the generation of electricity. In another study, Yang et al. [173] reported dye degradation with the use of a bio-electrical system. They concluded that microbes were used as oxidizing reagents to oxidize acetate at the anode. While Yadav et al. [174] reported that MFCs generate electricity during dye degradation. It was also reported that they have the highest efficiency of dye degradation, which is 96–100%. They also discussed the factors that can affect the production of electricity and the degradation of dye, which include bacteria, anodes, cathode membranes, cathode catalysts, and substrates. In another study, Yaqoob et al. [175] discussed the application of microbial fuel cells in remediation of environmental pollutants including dyes. 

There are many factors that may affect the effectiveness of any of the processes used for the abatement of dye removal, as shown in Table 2.

Table 3 presents the comparison of conventional versus advanced approaches and the advantages of recent strategies, respectively.

## 6. Challenges and Future Perspectives

Dye-containing wastewater is a major concern for the environment and human health. The most commonly used methods for color removal are chemical, physical, and biological treatments. The physicochemical methods are proven to be efficient, but they are costly. Thus, dye removal using biological methods is somewhat promising as a viable alternative to physicochemical methods. Nowadays, researchers are paying increased attention to advanced strategies of dye remediation. Microbial fuel cells have emerged as an essential and advantageous tool because they produce electricity for the remediation of pollutants and have a higher removal potential. Other approaches include nanotechnology, enzyme-mediated techniques, microbial remediation, phycoremediation, mycoremediation, and yeast-mediated remediation of dyes. The success rate and effectiveness of these methods are major challenges. The integration of these advanced methods may represent a complete solution for the abatement of dye pollution. Future studies should explore pure culture- as well as mixed culture-mediated decolorization processes in which microbial strains should be improved by utilizing genetic engineering. Further advanced research is required to study dye degradation pathways, the genetics of dye decolorizing microorganisms, and applications of recent dye treatment strategies. There is also a need to explore low-cost environmentally friendly treatment strategies for a sustainable environment.

## 7. Conclusions

Dye pollution is one of the major environmental problems and has negative effects on ecosystem health. As sustainable development is crucial globally, it is essential to develop a zero-waste approach, which is cost-effective too. There is a need to develop strategies for further dye remediation that are cooperative with the industries that are releasing dyes. Biological techniques can be integrated with advanced strategies to form a multifaceted, integrated, and cost-effective approach for improved dye remediation from wastewater. An appropriate environmentally friendly, simple and cost-effective strategy is needed for the efficient treatment of dye pollution. A biological approach using different microorganisms, nanotechnology, microbial fuel cells, and microbial biosorbents has great potential to remediate dyes for a cleaner and greener environment. The technical and economic viability of dye degradation methods should be critically studied. Future research is required to develop cost-effective and environmentally friendly novel biotechnological and innovative approaches for a cleaner and greener ecosystem.

## Figures and Tables

**Figure 1 toxics-11-00940-f001:**
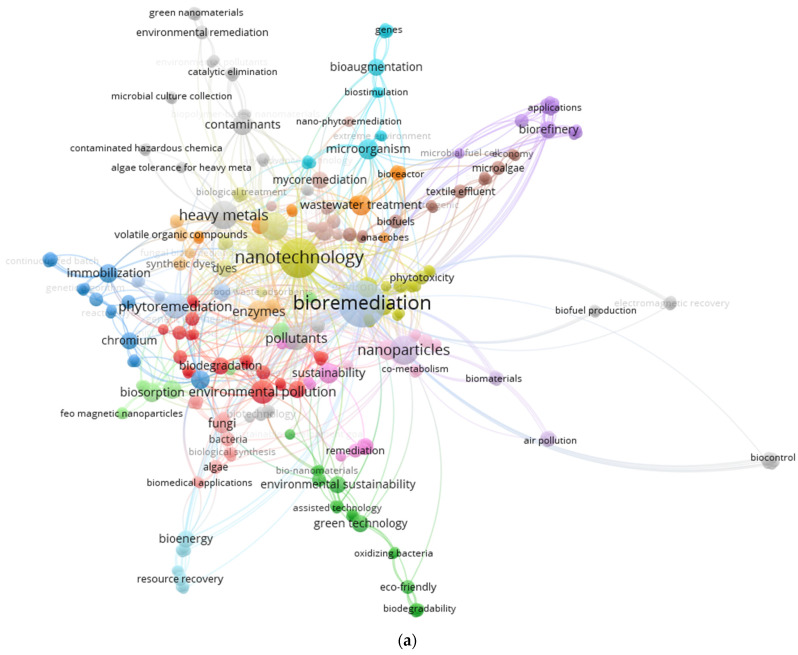
(**a**) Network visualization of bibliometric analysis of high-frequency keyword graphs over time and clustering of keyword citation networks. (**b**) Overlay visualization of bibliometric analysis of high-frequency keyword graphs over time and clustering of keyword citation networks. (**c**) Density visualization of bibliometric analysis. Term lines, circles, and other colors in a bibliometric network represent the degree of connectedness. The label and circle of an object grow in size in proportion to its weight. The size of the label and the matching circle that represents the object change in tandem with the item’s weight. A chronology is shown using a red-to-blue color gradient, with red designating earlier articles and blue representing more recent ones. This gradient acts as a visible counter for the passage of time. Red highlights important or noteworthy content, such as prominent citations, influential writings, or often-cited sources, while the more recent works, developing fads, or sources considered foundational in a certain discipline may all be represented by the color blue. Additionally, this may identify sources with a certain theme or content.

**Figure 2 toxics-11-00940-f002:**
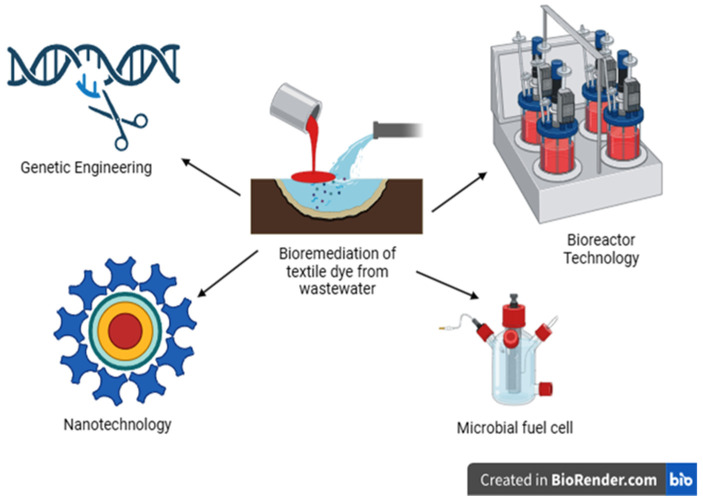
Recent possible strategies for dyes remediation from aquatic environment.

**Figure 3 toxics-11-00940-f003:**
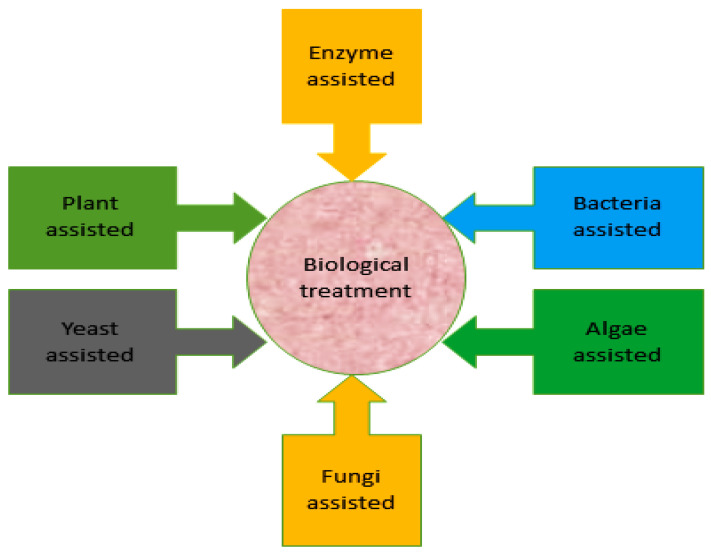
Different strategies for the bioremediation of dyes from polluted environments.

**Figure 4 toxics-11-00940-f004:**
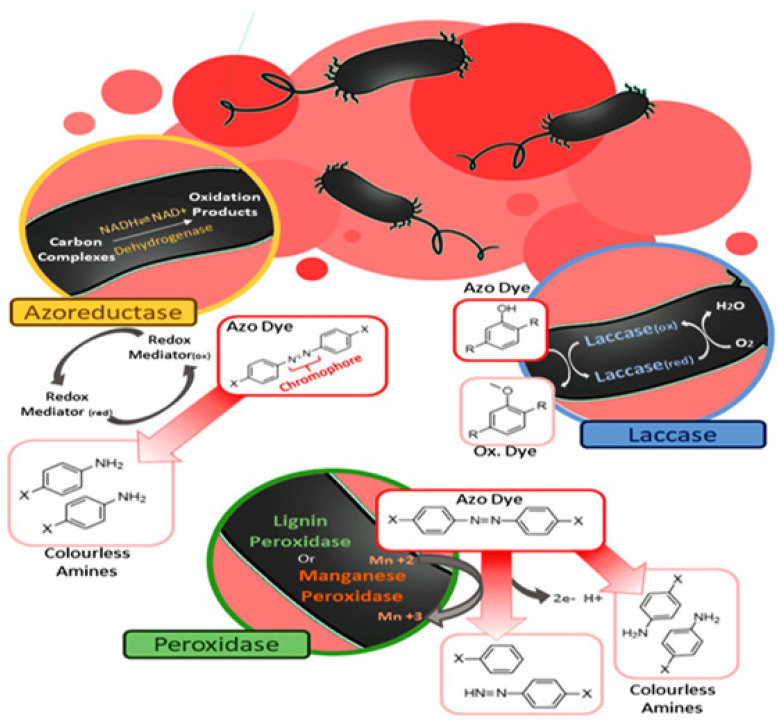
Schematic representation of three general bacterial enzymatic degradation mechanisms of the azo chromophore group, first showing the enzymatic degradation via the action of azoreductases—yellow—in this example using NADH as an essential reducing agent for the cleavage of azo bonds, generating aromatic amines and thus discoloring the medium. Then—clockwise—we have the catalytic reaction cycle mediated by laccase—blue—with the generation of an oxidized substrate instead of potentially toxic amines, in addition to not requiring cofactors. Finally, peroxidase enzymes—green—such as lignin peroxidase and manganese peroxidase, the two enzymes most commonly used for dye degradation, illustrate some possible products according to the cleavage of their bonds, which can be symmetric or asymmetric (Figure 4 is originally printed and adapted from Pinheiro et al. [87]. Copyright: © 2023 by the authors. Licensee: MDPI, Basel, Switzerland. This article is an open access article distributed under the terms and conditions of the Creative Commons Attribution (CC BY) license.

**Table 1 toxics-11-00940-t001:** Bacteria applied in the biodecolorization of different dyes.

Bacteria	Dye	% Decolorization	Reference
*Oerskovia paurometabola*	Acid red 14	91%	[114]
*Brevundimonas diminuta*	Rhodamine–B	90–95%	[115]
*Gerobacillus stereothermophilus* *ATCC 10149*	Remazol brilliant blue-R	90%	[116]
*Gerobacillus thermoleovorans*	Amaranth RI Fast red E	99%	[117]
*Serratia* spp.	Malachite green Crystal violet	96.5%	[107]
*Iodidimonas* spp.	Cationic dyes (Malachite green, crystal violet, methylene blue)	>90%	[118]
*Lysinibacillus sphaericus*	Reactive Yellow F3R Joyfix Red RB	96.30% 92.71%	[106]

**Table 2 toxics-11-00940-t002:** Different factors responsible for the advanced bioremediation of dyes.

S. No.	Factors	Advanced Oxidation Processes	Biological Treatment	Phytoremediation	Nanotechnology	Biotechnological Approaches
1.	Mechanism	Utilizes chemical oxidation to break down dyes and contaminants through highly reactive species (e.g., hydroxyl radicals).	Microorganisms, enzymes, and natural processes degrade dyes biologically.	Involves the use of plants to uptake, metabolize, or sequester dyes from the environment.	Utilizes nanomaterials or nanoparticles to adsorb, degrade, or facilitate the removal of dyes.	Relies on genetically modified or engineered microorganisms for enhanced dye degradation.
2.	Speed of Treatment	Generally rapid and efficient in dye degradation.	Biodegradation rates can vary and may be slower than chemical oxidation.	Treatment rates can be relatively slow, influenced by plant growth and environmental conditions.	Offers fast and efficient removal through high surface area and reactivity.	Can be engineered for rapid and targeted dye degradation.
3.	Selectivity	May not exhibit high selectivity and can degrade a wide range of dyes.	Microbes may exhibit selectivity towards specific dyes, affecting treatment effectiveness.	Plant species selection influences specificity, with variations in the range of dyes targeted.	Can be designed for selectivity through nanomaterial selection and modification.	Selectivity can be engineered by designing microorganisms with specific dye-degrading enzymes.
4.	Environmental Impact	May generate secondary byproducts and require careful management.	Generally environmentally friendly, with lower chemical use and reduced environmental impact.	Environmentally friendly, as it relies on natural processes and plant uptake.	Impact depends on nanomaterials used, with potential environmental concerns.	Environmental impact may vary depending on genetic modifications, but aims for minimal harm.
5.	Energy Requirements	Requires energy for chemical processes, potentially energy-intensive.	Typically energy-efficient, relying on microbial metabolism or natural processes.	Minimal energy requirements, as it mainly depends on plant growth.	Energy-efficient, but energy may be required for nanoparticle synthesis.	Energy-efficient, but genetic engineering may involve energy-intensive processes.
6.	Scale-Up Complexity	Scaling up AOPs can be complex and may require advanced infrastructure.	Scaling biological treatment systems is generally feasible but can be size-dependent.	Scalability can be challenging for large-scale phytoremediation projects due to space and time requirements.	Nanotechnology can be scaled up relatively easily but requires careful engineering.	Scalability depends on the cultivation and maintenance of engineered microorganisms, which can be challenging.
7.	Cost Effectiveness	Initial setup costs can be high due to equipment and chemical requirements.	Generally cost-effective in the long run due to lower operating costs and sustainability.	Cost-effectiveness depends on factors like plant species, maintenance, and project size.	Costs may vary depending on the nanomaterials used and their availability.	Costs can be higher due to research and development, genetic modification, and monitoring.
8.	Regulatory Considerations	May face regulatory scrutiny due to chemical usage and potential byproduct generation.	Typically meets regulatory compliance easily, especially for non-genetically modified organisms.	Regulatory approval depends on plant species used and potential ecological impacts.	Regulatory concerns related to nanoparticle release and toxicity may apply.	Requires regulatory approval for genetically modified organisms and potential ecological impacts.

Source: [176,177,178,179].

**Table 3 toxics-11-00940-t003:** Conventional vs. advanced dye bioremediation aspects.

S. No.	Aspects	Conventional Techniques	Advanced Techniques
1	Treatment Principle	Relies mainly on natural processes and microbial action.	Integrates innovative approaches and advanced materials.
2	Effectiveness	May be limited in the removal of complex and recalcitrant dyes.	Generally, more effective in breaking down a wide range of dyes.
3	Speed of Treatment	Biodegradation rates can be slow.	Often faster due to enhanced microbial activity and optimized conditions.
4	Microbial Strains	Utilizes naturally occurring microorganisms.	May involve the use of genetically modified or engineered microorganisms.
5	Nutrient Requirements	Requires standard nutrients and conditions for microbial growth.	May require tailored nutrient supplementation for specific dye degradation.
6	pH and Temperature Control	Typically relies on ambient conditions.	Requires precise control of pH and temperature for optimal performance.
7	Dye Specificity	Some microbes may exhibit selectivity towards certain dyes.	Can target a broader range of dyes through microbial diversity or modifications.
8	Toxic Byproducts	May produce secondary pollutants or byproducts.	Tends to generate fewer toxic byproducts due to targeted degradation.
9	Resilience to Shock Loads	Vulnerable to shock loads and fluctuations in dye concentrations.	Better equipped to handle variations in dye concentrations.
10	Scale-up Challenges	Scaling up conventional bioremediation processes can be challenging.	Advanced techniques may offer more scalability and adaptability.
11	Sustainability	Moderately sustainable, but environmental impacts may vary.	Aims for increased sustainability through optimized processes.
12	Costs	Typically lower initial costs but may require longer treatment times.	May have higher initial setup costs but can be more cost-effective in the long term.
13	Regulatory Compliance	Conventional methods may require fewer regulatory approvals.	Advanced techniques may face additional regulatory scrutiny due to genetic modifications or novel materials.
14	Treatment Principle	Relies mainly on natural processes and microbial action.	Integrates innovative approaches and advanced materials.
15	Effectiveness	May be limited in the removal of complex and recalcitrant dyes.	Generally, more effective in breaking down a wide range of dyes.
16	Speed of Treatment	Biodegradation rates can be slow.	Often faster due to enhanced microbial activity and optimized conditions.
17	Selectivity	Limited selectivity and may not effectively target specific dyes	Can be engineered for selectivity targeting specific dye pollutants

Source: [180,181,182,183,184,185].

## Data Availability

Not applicable.

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
