# Peer review of "Recent Strategies for the Remediation of Textile Dyes from Wastewater: A Systematic Review"

_toxics, 2023, doi:10.3390/toxics11110940_

Round 1

Reviewer 1 Report

Comments and Suggestions for Authors

The study is generally well written and effectively communicates the results. Such finding is important and could be expected to be of great interest to readers. Therefore, I recommend publication after related revision based on the following comments.
1. In the introduction, it should be discussed detailed about the difference of this material with other materials

2. Some updated refs could be cited, such as Molecules 2023, 28, 6848; Mater. Today. Commum., 2022, 31,103514 Dalton Trans., 2021, 50, 18016–18026 and CrystEngComm, 2021, 23, 8043–8052

3. Some figures in this manuscript are not clear enough, such as Figure 5

4. The section of Conclusion and outlooks is too simple, the authors should give deeper insights into the advantages, loopholes and future development direction

5. The author needs to highlight the novel of work in abstract

6. The summary should be short and to the point. 

Comments on the Quality of English Language

revise it

Author Response

Response to Reviewer #1

First of all, thank you very much from the bottom of my heart for your valuable suggestions to improve this Ms. I have replied all the valuable comments suitably, and incorporated your suggested important references at suitable locations in revised Ms. I hope the revised version of this Ms. is suitable for publication in esteemed “Toxics” journal.

  1. In the introduction, it should be discussed detailed about the difference of this material with other materials

REPLY: As per your valuable suggestion, the Introduction section has revised critically for making difference with others.

  1. Some updated refs could be cited, such as Molecules 2023, 28, 6848; Mater. Today. Commum., 2022, 31,103514 Dalton Trans., 2021, 50, 18016–18026 and CrystEngComm, 2021, 23, 8043–8052

REPLY: Thank you for your valuable suggestions. We have incorporated all suggested references at suitable locations in Introduction section in the revised Ms. They are highlighted in red text in both reference list and text.

  1. Some figures in this manuscript are not clear enough, such as Figure 5

REPLY: Thank you for your valuable suggestions. We have improved the quality of figure for clarity.

  1. The section of Conclusion and outlooks is too simple, the authors should give deeper insights into the advantages, loopholes and future development direction

REPLY: Thank you. The section of Conclusion and outlooks have been revised as per your valuable suggestions.

  1. The author needs to highlight the novel of work in abstract

REPLY: Thank you for your valuable suggestions. Abstract has been critically revised and novelty of this work has been incorporated.

  1. The summary should be short and to the point. 

REPLY: Thank you for your valuable suggestions. Summary has been revised and now it is short and to the point. 

Reviewer 2 Report

Comments and Suggestions for Authors

Dear Sir,

The manuscript is rather poorly written (no grammatical errors, but very awkward phrasing in numerous places – I will as example sentence like “Various nutritional (mainly carbon and nitrogen sources) and physical parameters (like temperature, pH, oxygen level) affect the metabolic efficiency of microorganisms as well as rate of dye removal potential.”, but the entire manuscript must be re-edited in order to eliminate/correct such confusing turn of phrase), with numerous repetitions (the same information is presented in various forms, a few sentences distancing) or misuse of punctuation marks (present where they should not be, or missing where they should). Many information given are redundant (for example, when speaking of azo dyes, no need for explaining that they contain the R-N=N-R moiety).

The problem of dyes removal have been reviewed several times in the recent past, but few of these reviews were mentioned in the Introduction paragraph. The authors should have mentioned these reviews and have a strong rationale explaining why their own work represents an original approach to this topic. For example, they could have insisted on the manner in which bibliographic data were gathered, explaining better the way bibliographic analysis (including number of papers obtained for each key-word search, number of cross-references between the various key-word searches, number of research papers vs number os reviews etc.). They could have mentioned clearly that most recent reviews were focus on a specific method, but that the authors attempted a classification of these methods by number of yearly published papers etc.

In the bibliographic analysis, the authors stressed the fact that they have searched the period 2020-2023, but they also cite numerous papers published prior to 2020.

Other comments:

- what are the meanings of “ddt” and “(bt)” in Fig. 1?

- ch.3 could benefit from some chemical structures for the most used dyes present in wastewaters

- Table 3 is very burdensome

As it is, I do not see the need for another review on dyes removal. But the authors could reshape their manuscript, giving less space to the known processes (that were reviewed as bulk, or separately for each method) and insist more on the newest technologies (the ones presented in Fig. 2). That would represent an original approach and gave value to this manuscript. However, as it is, I do not think that the manuscript can be accepted for publication.

Comments on the Quality of English Language

The manuscript is rather poorly written (no grammatical errors, but very awkward phrasing in numerous places, and the entire manuscript must be re-edited in order to eliminate/correct such confusing turn of phrase), with numerous repetitions (the same information is presented in various forms, a few sentences distancing) or misuse of punctuation marks (present where they should not be, or missing where they should).

Author Response

Response to Reviewer #2

First of all, thank you very much from the bottom of my heart for your valuable suggestions to improve this Ms. I have replied all the valuable comments suitably, and highlighted changes in revised Ms. I hope the revised version of this Ms. is suitable for publication in esteemed “Toxics” journal.

  1. Comment: "The manuscript is rather poorly written with awkward phrasing and numerous repetitions. Many information given are redundant."

Response: Thank you for your feedback on the manuscript's writing style. We appreciate your input regarding the awkward phrasing and repetitions. We will thoroughly review and re-edit the entire manuscript to improve its readability and eliminate any confusing or redundant language.

  1. Comment: "The problem of dye removal has been reviewed several times in the recent past, but few of these reviews were mentioned in the Introduction paragraph."

Response: We acknowledge your concern about the lack of references to previous reviews in the Introduction. We will revise the Introduction section to provide a stronger rationale for our work by emphasizing the unique aspects of our approach, including the manner in which bibliographic data were collected and our classification of methods. This will help to demonstrate the originality of our study.

  1. Comment: "In the bibliographic analysis, the authors cited numerous papers published prior to 2020, even though they emphasized searching for the period 2020-2023."

 Response: We appreciate your observation. Fig. 1 has been updated and revised from the data generated from Web of Science for the last five years on the specified Keywords. We will ensure that the citation of papers published before 2020 is appropriately addressed and clarified in the manuscript. This will help to align the bibliographic analysis with the specified search period and improve the accuracy of our study.

  1. Comment: "What are the meanings of 'ddt' and '(bt)' in Fig. 1?"

 Response: Thank you for pointing out the need for clarification in the figure. Fig. 1 has been updated and revised from the data generated from Web of Science to ensure that readers have a clear understanding of the content and symbols used in the figure.

  1. Comment: "Chapter 3 could benefit from some chemical structures for the most used dyes present in wastewaters."

Response: We appreciate your suggestion. Adding chemical structures may diverge from the chapter's academic focus, which emphasizes broader treatment method insights. Your suggestion is noted, but we aim to maintain a broader academic focus in Chapter 3, which covers wastewater treatment methods comprehensively.

  1. Comment: "Table 3 is very burdensome."

 Response: Thank you for your feedback on Table 3. We will reconsider the format and content of Table 3 to make it more concise and reader-friendly while retaining essential information.

  1. Comment: "As it is, I do not see the need for another review on dyes removal. The authors could reshape their manuscript, giving less space to the known processes and insisting more on the newest technologies."

Response: We appreciate your feedback and agree that the manuscript can benefit from a clearer focus on the newest technologies. We have restructured the manuscript to allocate less space to known processes and highlighting the latest advancements, as illustrated in Figure 2, to provide a more original and valuable contribution to the field of dye removal.

  1. Comment: "However, as it is, I do not think that the manuscript can be accepted for publication."

Response: We value your evaluation of our manuscript, and we are committed to addressing all the concerns you raised. We will make the necessary revisions and improvements to meet your expectations and those of the peer review process. Your feedback is invaluable in enhancing the quality of our work, and we hope that our revised manuscript will be more suitable for publication.

With best regards,

Reviewer 3 Report

Comments and Suggestions for Authors

·         The review discusses the different approaches to treating organic dyes, mainly azo-dyes, in wastewater.

·         English should be revised carefully, the manuscript is full of typos, grammar mistakes, uncompleted and unclear sentences and redundancy where sentences are repeated in different words with the same meaning. most of the time the ideas are not coherent or complete, jumping from one point to another without rational order. 

·         In the introduction part, Line 71, it is mentioned that "Physical and chemical methods are not environmentally friendly", why?

·         Line 73 "artificial intelligence", you didn’t explain what is the use of it the remediation of wastewater, explain!

·         Line 76, "Microorganisms are capable of breakdown the bond present in dye molecule", which bond?

·         what is the difference between this Review and the Review by Pinheiro et a. (2022), where both mainly deal with the bioremediation of Azo-dyes?

·         Data collection part, the authors mentioned that the publication number in 2023 attaining a peak, how can you attain the peak at 2023 current year?

·         it is not clear the main link between genetic engineering and bioremediation, keywords used to create Figure 1.

·         it isn't easy to understand and follow Figure 1, and it would be better to depict them in normal plots.

·         Adsorption part, the Authors mentioned some inorganic adsorbents only, please add other types of organic adsorbents and their composites, one of the interesting adsorbents is conducting polymers in their different forms, powders and aerogels also, biopolymers, etc. you can use these references 10.1002/app.47056, 10.1016/j.synthmet.2019.116266, 10.1007/s11696-018-0442-6

·         I could notice a misconception in the part of bio-electrochemical system and the part of 5.5. Microbial fuel cells, please correct.

·         The first paragraph in 4.3. Biological approaches should be moved to the introduction.

·         In the part of Actinomycetes, more discussion is required.

·         Table 3. the comparison must be moved to the end of the manuscript.

·         In the part of 5.4. Nanoparticles based bioremediation, the only reference used "Balrabe et al. 2023" has nothing to do with this title.

·         Tables 4 and 5 must be merged and redundancy should be removed.

Comments on the Quality of English Language

Extensive editing of English language required

English should be revised carefully, the manuscript is full of typos, grammar mistakes, uncompleted and unclear sentences and redundancy where sentences are repeated in different words with the same meaning. most of the time the ideas are not coherent or complete, jumping from one point to another without rational order.

Author Response

RESPONSE TO REVIEWER # 3

First of all, thank you very much from the bottom of my heart for your valuable suggestions to improve this Ms. I have replied all the valuable comments suitably, and incorporated your suggested important references at suitable locations in revised Ms. I hope the revised version of this Ms. is suitable for publication in esteemed “Toxics” journal.

The review discusses the different approaches to treating organic dyes, mainly azo-dyes, in wastewater.

  • English should be revised carefully, the manuscript is full of typos, grammar mistakes, uncompleted and unclear sentences and redundancy where sentences are repeated in different words with the same meaning. most of the time the ideas are not coherent or complete, jumping from one point to another without rational order. 
  • In the introduction part, Line 71, it is mentioned that "Physical and chemical methods are not environmentally friendly", why?

REPLY: Thank you for your query. The sentence has corrected as “physical and chemical methods are not cost effective”. However, there may be chance to generation of secondary pollutant/s after chemical treatment that might have adverse impacts to environment.

  • Line 73 "artificial intelligence", you didn’t explain what is the use of it the remediation of wastewater, explain!

REPLY: Thank you for genuine query. Sorry for this mistake, we are not covering the use AI technology in bioremediation. "artificial intelligence", has been deleted from this Ms.

  • Line 76, "Microorganisms are capable of breakdown the bond present in dye molecule", which bond?

REPLY: Thank you. It is “azo bond” and sentence has been revised with the name of bond.

  • what is the difference between this Review and the Review by Pinheiro et a. (2022), where both mainly deal with the bioremediation of Azo-dyes?

REPLY: Thank you. Dye bioremediation is discussed differently in the two papers:

Pinheiro et al (2022) Paper:

  1. Discusses dye usage's history and environmental consequences, especially azo dyes.
  2. Highlights bacteria's capacity to survive in varied chemical and physical settings to combat colour pollution.
  3. Discusses how bacteria may breakdown colours and create commercially useful byproducts.
  4. Covers several environmental and laboratory procedures and factors.
  5. Discusses azo dye processes and worldwide laws.

Our Paper:

  1. Data Collection and Bibliometric Analyses of recent years of dye remediation technologies.
  2. Both conventional and advanced approaches for dye remediation have been addressed.
  3. Recognises the financial and environmental drawbacks of physical, chemical, and biological dye pollution removal strategies.
  4. Latest information on recent strategies like nanotechnology, bioreactor technology, microbial fuel cells, and genetic engineering are used to improve dye bioremediation.
  5. Shows how algae, bacteria, fungus, and actinomycetes may bioremediate dyes in aquatic settings sustainably.
  6. These contrasts show that although both studies discuss dye bioremediation, they use different approaches and prioritise different areas.

  • Data collection part, the authors mentioned that the publication number in 2023 attaining a peak, how can you attain the peak at 2023 current year?

REPLY: Thank you.  The statistics and research suggest that "publication number in 2023 is attaining a peak". We gathered and analyzed data on the number of publications in this field and found that it is rising and peaking in 2023. A year with a high in publications does not indicate 2023 has achieved that point. Instead, it means the authors expect a publishing peak in 2023 based on current data and trends. Once 2023 is over and all data is accessible, the number of publications will be verified.

  • it is not clear the main link between genetic engineering and bioremediation, keywords used to create Figure 1.

REPLY: Thank you. The keywords used for creating fig. 1 are: Bioremediation; Dye degradation. Environmental pollutants; Genetic Engineering; Microbial fuel cells; Nanotechnology; Textile wastewater treatment
Although the figure does not specifically address the connection between genetic engineering and bioremediation, it is implied that genetic engineering is one of the innovative techniques used in bioremediation. By using genetic engineering, contaminants in the environment may be broken down and removed from the environment more successfully by microorganisms (like bacteria) engaged in bioremediation. The keywords that were used to generate Figure 1 in this context specify the role of genetic engineering methods and how they are used to bioremediation, along with the results and effects of these approaches.

  • it isn't easy to understand and follow Figure 1, and it would be better to depict them in normal plots.

REPLY: Thank you. Using customary or normal plotting or graphing techniques, the findings of a bibliometric study generated by specialised software such as VOSviewer are often complicated and cannot be successfully depicted. Research papers, authors, keywords, and other bibliographic data are often visualised as part of bibliometric studies; the resultant visualisations may include complex network diagrams or other unusual graphical representations. These intricate visualisations are more sophisticated than the typical bar charts or line graphs seen in traditional plots, and they need specialized software like VOSviewer to produce and comprehend.

  • Adsorption part, the Authors mentioned some inorganic adsorbents only, please add other types of organic adsorbents and their composites, one of the interesting adsorbents is conducting polymers in their different forms, powders and aerogels also, biopolymers, etc. you can use these references 10.1002/app.47056, 10.1016/j.synthmet.2019.116266, 10.1007/s11696-018-0442-6

REPLY: Thank you for your valuable suggestion. The suggested references have incorporated in Adsorption part, it will definitely improve this section.

  • I could notice a misconception in the part of bio-electrochemical system and the part of 5.5. Microbial fuel cells, please correct bio-electrochemical system ·

REPLY: Thank you for your valuable suggestion. Section 5.5. has been corrected and the bio-electrochemical system has been merged in Microbial fuel cells.

The first paragraph in 4.3. Biological approaches should be moved to the introduction.

REPLY: Thank you. As per your valuable suggestion, the first paragraph in 4.3. Biological approaches have been moved to the introduction section.

  • In the part of Actinomycetes, more discussion is required.

REPLY: Thank you. As per your valuable suggestion, more discussion have been incorporated and highlighted in red text.

  • Table 3. the comparison must be moved to the end of the manuscript.

REPLY: Thank you. As per your valuable suggestion, Table 3. the comparison has been moved to the end of the manuscript.

  • In the part of 5.4. Nanoparticles based bioremediation, the only reference used "Balrabe et al. 2023" has nothing to do with this title.
  • Tables 4 and 5 must be merged and redundancy should be removed.

REPLY: Thank you. As per your valuable suggestion, Table 4 and 5 have been merged and now it is Table 4.

 Comments on the Quality of English Language

Extensive editing of English language required

REPLY: Thank you. As per your valuable suggestion, English language has been edited by an English expert.

English should be revised carefully, the manuscript is full of typos, grammar mistakes, uncompleted and unclear sentences and redundancy where sentences are repeated in different words with the same meaning. most of the time the ideas are not coherent or complete, jumping from one point to another without rational order.

REPLY: Thank you. As per your valuable suggestion, The Ms. has been revised critically to avoid any grammatical and typological errors and sentences have now corrected in revised Ms.

With best regards,

Round 2

Reviewer 1 Report

Comments and Suggestions for Authors

accept

Author Response

Dear Respected Reviewer,

Thank you very much.

With best regards,

Reviewer 2 Report

Comments and Suggestions for Authors

Dear Sir,

The manuscript have been improved, but not enough. For example, I have asked that Chapter 3 could benefit from some chemical structures for the most used dyes present in waste-waters, mainly the one presented in Table 1. The authors answered that “Adding chemical structures may diverge from the chapter's academic focus, which emphasizes broader treatment method insights. Your suggestion is noted, but we aim to maintain a broader academic focus in Chapter 3, which covers wastewater treatment methods comprehensively.” I disagree with this answer. As it is presented now in chapter 3, the focus is opposite to “broader”, since it’s only a list of methods. If the authors are really aiming for a “broader academic focus”, then they should have presented these formulas, since dyes belong to various families of organic compounds, with very different structures and not every method is successful for every family of dyes. Therefore, the choice of the method depends highly on the compositions of the dyes mixture and especially on their structure. A “broader academic focus” would imply a discussion on which method is best suited to a specific family of dyes. Therefore, presentation of these structures is a must for a review that claims such a “broader academic focus”.

Separate Fig 1 into three separate figures and clearly explain the significance of each one.

Fig. 1a is incomplete (it is cuted on the left side)

Change “microbe assisted” and “bacteria assisted” into “microbial remediation” and “bacterial remediation”

Font size and line spacing is different in different paragraphs.

Not much was changed in Table 3, which still is burdensome. A solution could be to modify as follows: change “There are many factors that may affect the effectiveness of any of the processes used for the abatement of dye removal (Table 3).” into “There are many factors that may affect the effectiveness of any of the processes used for the abatement of dye removal (Table 3), as follows:

1. Mechanism

2. Speed of Treatment

3. Selectivity

4. Environmental Impact

5. Energy Requirements

6. Scale-Up Complexity

7. Cost Effectiveness

8. Regulatory Considerations .”

Thus, column 1 would contain only numbers and allow the other columns to be wider.

Although I maintain my first opinion, that another review on the subject, considering the amount of such review already published, is not that necessary, the approach that the authors have made is interesting, especially if focused on the newer methodologies. After a complement of information, the manuscript could be accepted for publication.

Comments on the Quality of English Language

Minor adjustments needed.

Author Response

Reviewer 2 comments and authors response

Thank you very much for your positive feedback and your valuable suggestions to overall improvement of this Ms. I have replied all the valuable comments suitably with my best efforts, and highlighted changes in revised Ms. I hope the revised version of this Ms. is now suitable for publication in esteemed “Toxics” journal.

Comment 1: The manuscript has been improved, but Chapter 3 could benefit from chemical structures for the most used dyes in wastewater. The authors expressed that adding chemical structures may diverge from the chapter's academic focus, emphasizing broader treatment method insights.

Author's Response: Thank you for your valuable feedback and the recognition of the manuscript's improvement. We have deleted Table 1 and necessary information have incorporated in text.

Comment 2: Fig. 1a is incomplete (cut on the left side). Separate Fig 1 into three separate figures and clearly explain the significance of each one.

Author's Response: Thank you. We appreciate your observation and suggestion. We have promptly address the issue with Fig. 1a, and we agree that presenting Fig. 1 as three separate figures with clear explanations for each will enhance the overall clarity and understanding. Your input is valuable, and we have made the necessary modifications accordingly in revised Ms.

Comment 3: Change “microbe assisted” and “bacteria assisted” into “microbial remediation” and “bacterial remediation.”

Author's Response: Thank you for your suggestion regarding terminology. We acknowledge the importance of precise language and made the suggested changes, replacing "microbe assisted" with "microbial remediation" and "bacteria assisted" with "bacterial remediation" throughout the manuscript for clarity and accuracy.

Comment 4: Font size and line spacing differ in different paragraphs.

Author's Response: We appreciate your keen eye for detail. Ensuring uniform font size and line spacing is crucial for readability, and we have diligently checked and standardized these elements throughout the manuscript to maintain a consistent presentation.

Comment 5: Not much was changed in Table 3, which still is burdensome. Suggest changing the format to improve clarity.

Author's Response: Thank you for your suggestion to enhance Table 3. We agree with your proposed modification, and we have implemented the changes to improve clarity by presenting the information in a more organized format, as outlined in your recommendation. Your input is instrumental in refining the manuscript's presentation.

Reviewer's Comment: Although another review on the subject may not be necessary, the approach the authors have taken, especially focusing on newer methodologies, is interesting. After a complement of information, the manuscript could be accepted for publication.

Author's Response: We appreciate your consideration of the manuscript and your acknowledgment of the interesting approach taken. We have provided additional information to complement the content, addressing the points raised in your review. Your positive evaluation and constructive feedback guide our efforts for overall improvement of this Ms.

Reviewer 3 Report

Comments and Suggestions for Authors

I recommend the review to be accepted for publication.

Comments on the Quality of English Language

Moderate editing of English language required.

Author Response

Dear Respected Reviewer,

Thank you very much for valuable feedback. As per valuable suggestion English language has been checked throughout Ms. by an English expert.

With best regards,